# Origami-Inspired Structure with Pneumatic-Induced Variable Stiffness for Multi-DOF Force-Sensing

**DOI:** 10.3390/s22145370

**Published:** 2022-07-19

**Authors:** Wenchao Yue, Jiaming Qi, Xiao Song, Shicheng Fan, Giancarlo Fortino, Chia-Hung Chen, Chenjie Xu, Hongliang Ren

**Affiliations:** 1Department of Electronic Engineering, The Chinese University of Hong Kong, Hong Kong, China; wenchao.yue@link.cuhk.edu.hk; 2Department of Biomedical Engineering, National University of Singapore, Singapore 119077, Singapore; qi.jiaming@u.nus.edu (J.Q.); e0546097@u.nus.edu (X.S.); e0546018@u.nus.edu (S.F.); 3Shenzhen Research Institute, The Chinese University of Hong Kong, Shenzhen 518172, China; 4Department of Computer Science, University of Calabria, 87036 Rende, Italy; g.fortino@ieee.org; 5Department of Biomedical Engineering, City University of Hong Kong, Kowloon, Hong Kong, China; chiachen@cityu.edu.hk (C.-H.C.); chenjie.xu@cityu.edu.hk (C.X.); 6Shun Hing Institute of Advanced Engineering, The Chinese University of Hong Kong, Hong Kong, China

**Keywords:** origami-inspired, variable stiffness, force-sensing, microfiber, pneumatic actuation

## Abstract

With the emerging need for human–machine interactions, multi-modal sensory interaction is gradually pursued rather than satisfying common perception forms (visual or auditory), so developing flexible, adaptive, and stiffness-variable force-sensing devices is the key to further promoting human–machine fusion. However, current sensor sensitivity is fixed and nonadjustable after fabrication, limiting further development. To solve this problem, we propose an origami-inspired structure to achieve multiple degrees of freedom (DoFs) motions with variable stiffness for force-sensing, which combines the ductility and flexibility of origami structures. In combination with the pneumatic actuation, the structure can achieve and adapt the compression, pitch, roll, diagonal, and array motions (five motion modes), which significantly increase the force adaptability and sensing diversity. To achieve closed-loop control and avoid excessive gas injection, the ultra-flexible microfiber sensor is designed and seamlessly embedded with an approximately linear sensitivity of ∼0.35 Ω/kPa at a relative pressure of 0–100 kPa, and an exponential sensitivity at a relative pressure of 100–350 kPa, which can render this device capable of working under various conditions. The final calibration experiment demonstrates that the pre-pressure value can affect the sensor’s sensitivity. With the increasing pre-pressure of 65–95 kPa, the average sensitivity curve shifts rightwards around 9 N intervals, which highly increases the force-sensing capability towards the range of 0–2 N. When the pre-pressure is at the relatively extreme air pressure of 100 kPa, the force sensitivity value is around 11.6 Ω/N. Therefore, our proposed design (which has a low fabrication cost, high integration level, and a suitable sensing range) shows great potential for applications in flexible force-sensing development.

## 1. Introduction

Human–machine integration involves a deep interaction among humans, machines, and environments. Tactile (or force information) plays an indispensable role as one of the indispensable media, together with visual and auditory signals [1,2,3]. It is well known that humans directly interact with the environment through tactile or force perceptions through hands or fingers [4]. While in robot-assisted minimally invasive surgery (RMIS), doctors remotely operate a surgical robot to achieve minimally invasive surgery (MIS), and the accurate tactile or force-sensing guarantees the precise operation [5,6]. Therefore, force-sensing devices have become essential interfaces for the collaboration between humans and medical robots [7].

According to the mounting location, force-sensing devices can be divided into two categories: one is mounted on the ground or the desktop, and the other is mounted on the human body [8,9]. The force-sensing devices mounted on the ground (such as the commercial device Phantom Omni) can perceive multiple degree-of-freedom (DoF) force signals, but they usually have poor mobility [10]. Meanwhile, the setup of force sensors is relatively fixed, resulting in such force-sensing devices perceiving force within a relatively limited region. Wearable force sensors are attached to the fingers or human hands, leaving the hands in an occupied state, thus interfering with the user’s operation and reducing the immersion during the operation procedure [11]. Wearable devices mounted on the other part of the body, such as the head, can free hands. Still, regarding the part where such a device is mounted, it is hard to apply external force to permit passive tactile or force feedback [12]. Therefore, to solve the problem above, researchers have paid more attention to forearm-mounted or twist-mounted methods as alternative ways to collect force-sensing information from the environment without sacrificing the freedom of the hand [13].

Most traditional pressure or force sensors are made from rigid substrates, such as silicon and metal [14]. Due to their rigid and brittle nature, such materials will be hard to conformally contact with the surface of human skin for collecting accurate data. For example, force-sensing resistors are good products for force measurements. Although they are slim and flexible, they are still relatively rigid [15]. Therefore, the interest in developing flexible sensing systems has rapidly received significant attention over the last decade. Origami is a traditional form of art that dates back hundreds of years [16]. Due to their lightweight characteristics and stiff-less structures, origami design has recently obtained a surge of interest, while reconfigurable capabilities allow them to achieve more flexibility [17,18,19]. Those advantages benefit the development of wearable devices [20,21,22] as they can be adaptable, compact, and lightweight [23]. In the scene of interaction with the skin, we can classify origami structures into circular type and linear type, according to the pressure direction. There are prototypes developed for other applications that are inspiring. The Miura-Ori pattern may be the most widely used structure because of the simplicity of fabrication, structural variations, and functional characteristics [24]. Miura-Ori origami structures contain deformable double corrugation surfaces [25], which can contract and extend simultaneously in two axes. Pooya et al. proposed a cyclic origami structure for rotorcraft collision protection, which is a variation of the classic Miura-Ori pattern [26]. The cyclic structure can contract and extend radially under actuation. The origami structure can also be fabricated as an artificial muscle. When the wavy origami skeleton is sealed in PVC films, the FOAM can perform a large pulling force actuated with negative pressure [27]. For the linear structure, Matthew et al. presented a “pneumagami” module, whose origami structures are fabricated in multiple layers, including rigid bodies, flexible hinges, and control circuits [28]. This module can achieve a bending motion in two axes and vertical extension actuated by soft pneumatic actuators with three degrees of freedom.

Therefore, based on the investigation above, our work proposes the origami-inspired structure for force-sensing, which can be potentially utilized as a wearable device for collecting human motion signals, as shown in Figure 1a. The foldable structure can be fixed to the strap, thus making it easily wearable on the wrist. In the uninflated state, the user can adjust the fixing position of the foldable structure on the wrist at will; in the inflated state, the overall stiffness of the foldable mechanism could change with the air injection process: on the one hand, it can be inflated until achieving a comfortable wearing volume, which can achieve the fixation function; on the other hand, the sensitivity of the force-sensing system can be adjusted to a suitable sensing zone by adjusting the pressure of the inflation air. To summarize, the main contributions and novel points in the work can be summarized as follows:(1)A low-cost, easy-to-use, origami-inspired force-sensing device with a sandwich structure is proposed; the utilization of common Eco-Flex 50, polydimethylsiloxane (PDMS), and cardboard provide the structure with multi-DOF motions to sense various types of forces.(2)A microfiber sensor with a double-layer structure was designed to improve sensitivity by eliminating strain at the edges. It was seamlessly integrated into the origami-inspired system through a multilayer casting technique.(3)A pneumatically variable sensing system sensitivity is introduced to solve the traditional problem of fixed sensor sensitivity and further expand the adaptability of sensors to different application scenarios.

## 2. Materials and Methods

This paper integrates the origami-inspired structure (made of ordinary cardboard) with an ultra-flexible microfiber sensor to create a multi-DOF force-sensing system with variable stiffness. The subsequent sub-section covers the design and fabrication process of the origami structure and microfiber sensors, followed by the integration between the origami structure and microfiber.

### 2.1. Design of Origami Structure

Origami structures, consisting of rigid parts and flexible hinges, have the potential to sustain large force renderings without buckling while being conformal to the skin at the same time. Other advantages, such as low cost, lightweight, and a simple fabrication process, facilitate the development of wearable origami technologies. The structure was folded from a hexahedron with a size of 60×60×50 mm. The folding pattern of this origami model is shown in Figure 1b. Figure 1c shows the pattern design of this foldable structure containing three different patterns, which constrained the system with only one degree of freedom allowing the structure to move linearly along the *Z* direction. However, aiming to increase dexterity and achieve multiple actuation modes, the whole origami structure was built on the commonly used cardboard. Meanwhile, the middle points of ceases were all processed with diamond cutouts to remove the fixed distance constraint shown in Figure 1c. Such diamond cutouts can also release the stress concentration of foldable structures to increase their mechanical cycle lives [29]. The gaps amongst connections are preferably glued using Universal ERGO^®^ 5011 to compact the structure. The glue is used to ensure the airtightness of the force-sensing system.Using the glue ensures no air leaks at the bonding locations and achieves the stiffness maintenance of the whole sensing system. In this paper, the glue ERGO^®^ 5011 is a robust, fast-drying glue from the Swiss company ergo, and is suitable for silicone bonding. The glue ERGO^®^ 5011 is also very efficient, taking only about 10–20 s to dry and position, ensuring the connection precision of the crease joints. To achieve conformal contact and even pressure on the wrist, the green square surface contacting the skin shown in Figure 1a was fabricated out of silicone rubber (Ecoflex 00–50). To reduce the complexity of the origami structure, each piece of the segment was designed in a sandwich configuration. The rigid cardboards were wrapped inside, and the thinner Eco-Flex layer at the junction served as the flexible hinge when folded. A microfiber pressure sensor was embedded in the Eco-Flex surface to monitor the applied pressure to achieve closed-loop control.

### 2.2. Design of Ultra-Flexible Microfiber Sensor

In this section, the proposed ultra-flexible microfiber sensor can adhere to the contact surface and easily integrate into our origami structure above, to achieve closed-loop control and sensitive pressure awareness. This design (in our work) was adopted mainly from the following two perspectives. Considering the excellent reconfigurability of the microfiber sensor in our scenario, the substrate and microfiber sensor were two separate parts. We only needed to build substrate channels with different shapes for mounting the microfiber sensor, simplifying the fabrication and design process. Meanwhile, most microfluidic or microchannel sensors were fabricated with a sandwich structure, which weakened the deformation of the inner sensing element, leading to a lower sensitivity. Therefore, our pressure sensor was composed of PDMS and liquid metal, where PDMS functioned as the substrate material and liquid metal functions as the sensitive element. The design of the strain sensor was derived from the conventional wire-type strain sensor, with liquid metal embedded in PDMS in the form of a microfluidic channel. Both the base material and the sensitive element have excellent stretchability, and the sensor designed in this paper mainly adopted a strain-gate structure. This combination increased the total length of the microfluidic channel, improving the sensor’s resistance and sensitivity. The diameter of the substrate channel was adjustable at around 0.5 mm, and the total length was around 20 mm. As for the size of the microfiber sensor, we could control the diameter of the footprint to become adjustable from 0.5–0.6 mm.

### 2.3. Fabrication Process

Based on the previous design section, this section executes the fabrication and assembly of the origami model and microfiber sensor. We conducted the feasibility tests for our fabrication to evaluate the sensing capacity and origami kinematics. The details are in the following subsections.

#### 2.3.1. Microfiber Sensor Fabrication

To achieve sensitive pressure awareness and a high reconfigurable structure, we leveraged liquid metal (eGaIn) to fill a hollow fiber structure. According to the reported reference [30], we opted for the extrusion technique as the facile fabrication method for microfiber. Briefly speaking, a metal filament (controllable size) was used to vertically draw from a lab syringe in which the prepared polydimethylsiloxane (PDMS) base had been mixed with a curing agent at a ratio of 10:1 (followed by the degassing process). Since PDMS has excellent viscoelastic properties and high surface tension, it will form a uniform thin layer on the surface of the metal filament during the drawing process.This step follows the metal filament with uncured PDMS, which will pass through an electrical heater, placed vertically above the syringe to complete the final curing step for PDMS. After coating the filament with cured PDMS, the wire is extruded from the PDMS out layer.

As such, it will form an elastomeric hollow fiber structure. At this time, the tubular structure is injected by the liquid metal eGaIn to form the conductive channel. To avoid the leakage of eGaIn, the epoxy encapsulates both sides of the microfiber. Meanwhile, two electrical cables are inserted to serve as outlet connectors. As shown in Figure 2a, the microfiber sensor prototype is under a manually stretching test, and the digital multimeter is utilized to record the change of resistance during the test. The basic layout of the microfiber sensor with a substrate channel is presented in Figure 2b.

#### 2.3.2. Origami Fabrication

The origami is designed with a sandwich structure to render lightweight and fabrication simplicity. The rigid origami plates and Eco-Flex mold is 3D-printed after modeling in SolidWorks.The degassed Eco-Flex precursor mixture is poured into the mold as the first layer, then put in the oven for one hour with a temperature of 70 ${}^{\circ }$C. After fully cured, the cardboards are placed on the Eco-Flex according to the pattern shown in Figure 1c. Then, the second layer of Eco-Flex is poured into the mold, wrapping the cardboard inside. The cured Eco-Flex is de-molded, and the flexible pressure sensor is placed in a pre-designed location to monitor real-time pressure. Finally, the origami is folded, and the junction between connecting cardboard is glued to form a 3D configuration. Meanwhile, the gaps are sealed using PDMS films. Figure 3 shows that the assembled device can generate the passively linear motion against the table when applied to external pressure.

#### 2.3.3. Integration Fabrication

After fabrication and verifying the feasibility of the sensor and the origami structure, more details on the integration procedure will be presented in this subsection. The integration process is mainly divided into two steps: the preparation process and the fabrication process. In the preparation process, the specific mold is designed using the model software SolidWorks and fabricated through 3D printing. At the same time, the Eco-Flex 00-50 silicone solutions 1A and 1B (ratio 1:1) are mixed thoroughly for 2–3 min, and the mixtures are set aside under the cool condition. After finishing the preparation stage, the silicone mixture filled half of the mold’s depth for making the outer layer. Then the mold with a half mixture solution will be put into the vacuum thermostat (within 70 degrees Celsius) for about an hour. The next step is manually putting the creased cardboard into the mold and compact pattern layer. After checking that the surface fits, the rest of the silicone mixture solution will be poured into the mold until filled to make the inner layer. The mold will be placed in a vacuum oven (within 80 degrees Celsius) for about half an hour. During the final curing process, the pre-prepared microfiber sensor is laminated to the inner layer of the cured foldable structure. The installation timing can be chosen when the Eco-Flex layer is not fully cured, or when the inner layer has good adhesion. The finished model is folded according to the previous folds, and the joints are glued with universal ERGO^®^ 5011. The whole system is sealed with PDMS film to ensure airtightness. The integration procedure is well illustrated in Figure 4.

## 3. Results

This section discusses the kinematic analysis and simulation study for this proposed origami structure. The performance tests for our microfiber sensor were operated to evaluate the sensing stability and feasibility. Meanwhile, the calibration experiment was also operated to build the relationship between resistance and applied pressure. Moreover, the online acquisition experiment was conducted to verify the pressure generating and sensing capability. Finally, the variable stiffness experiment was conducted to prove that the pneumatic actuation can cause the variable stiffness of the structure to affect the sensitivity of the sensor. The details will be further given in the following subsections.

### 3.1. Origami Structure Analysis and Experiment Test

#### 3.1.1. Kinematic Analysis of Origami Model

Most force-rendering display device kinematics remain unchanged when mounted on the body as they consist of rigid segments and joints [31,32]. To achieve the various modes of force rendering displays, it is necessary to understand the kinematic model of our foldable structure. This paper uses the top surface as a contact plane to derive its kinematics equations.

The foldable structure is modeled as two square planes with the side length of *L* shown in Figure 5. Assuming that the bottom plane is connected with the fixation belt, it can be regarded as fixed, while the top plane is free of external constraints. To better describe the coordinate system, we choose the center points $O_{1}$ and $O_{2}$ of both planes as the coordinate origins, and we define the unit vector $\vec{n}=\left(x_{0},y_{0},z_{0}\right)$ as the normal vector of the top plane. Based on the definition above, the relative Cardan angles $\delta_{r}$ and $\delta_{p}$ generated from the roll motion and pitch motion can be derived from the following equations:(1)$$\delta_{r}=\arccos\frac{z_{0}}{\sqrt{x_{0}^{2}+z_{0}^{2}}}$$
(2)$$\delta_{p}=\arccos\frac{z_{0}}{\sqrt{y_{0}^{2}+z_{0}^{2}}}$$

In our scenario, we choose to actuate this origami module by the pneumatic method described previously. Due that the pneumatic channels being distributed symmetrically, the effects of the resultant force renderings are to drive the four legs of the origami model separately for generating motions. The acting direction of the resultant force rendering can be equivalently along the $\vec{M_{1}^{\prime }M_{1}}$ and $\vec{M_{2}^{\prime }M_{2}}$, which are vectors connected with the mid-points of the corresponding sides:(3)$$\vec{M_{1}^{\prime }M_{1}}=\left[\begin{array}{c} 0\\ \frac{L}{2}\cdot \left(1-\cos\left(\delta_{r}\right)\right)\\ {M_{2}^{\prime }M_{2}}_{z}+\frac{L}{2}\cdot \sin\left(\delta_{r}\right) \end{array}\right]$$
(4)$$\vec{M_{2}^{\prime }M_{2}}=\left[\begin{array}{c} \frac{L}{2}\cdot \left(1-\cos\left(\delta_{p}\right)\right)\\ 0\\ \;{M_{1}^{\prime }M_{1}}_{z}+\frac{L}{2}\cdot \sin\left(\delta_{p}\right) \end{array}\right]$$

When both planes form a certain angle as roll or pitch motion occurs, there always exists a virtual ball tangent to two planes through point $O_{1}$ in the bottom plane and point $O_{2}^{\prime }$ in the bottom plane; its radius is defined as *r*. Here, we further define that $\vec{O_{1}O_{2}^{\prime }}={\left(x_{1},y_{1},z_{1}\right)}^{⊤}$. Therefore, we can further derive from the coordination origin $O_{1}$ to have the following equations,
(5)$$\vec{O_{1}O_{2}^{\prime }}=\left[\begin{array}{c} x_{1}\\ y_{1}\\ z_{1} \end{array}\right]=r\cdot \left[\begin{array}{c} 0\\ 0\\ 1 \end{array}\right]+r\cdot \left[\begin{array}{c} x_{0}\\ y_{0}\\ z_{0} \end{array}\right]$$ Furthermore, by combining (3) and (4), the transformation equation between the coordinate origins $O_{1}$ and $O_{2}$ is built as follows:(6)$$\begin{array}{cc} \; & \vec{O_{1}O_{2}}=\frac{\vec{O_{1}M_{1}}+\vec{O_{1}M_{2}}}{2}+\vec{n}\times \frac{\vec{O_{1}M_{2}}-\vec{O_{1}M_{1}}}{2}\\ \; & =\left[\begin{array}{c} \frac{L}{2}-\frac{L}{4}\cdot \cos\left(\delta_{p}\right)\\ \frac{L}{2}-\frac{L}{4}\cdot \cos\left(\delta_{r}\right)\\ \frac{M_{2}^{\prime }M_{2_{z}}+M_{1}^{\prime }M_{1_{z}}}{2}+\frac{L}{4}\cdot \left(\sin\left(\delta_{p}\right)+\sin\left(\delta_{r}\right)\right) \end{array}\right]\\ + & \left[\begin{array}{c} x_{0}\\ y_{0}\\ z_{0} \end{array}\right]\times \left[\begin{array}{c} -\frac{L}{4}\cdot \cos\left(\delta_{p}\right)\\ \frac{L}{4}\cdot \cos\left(\delta_{p}\right)\\ \frac{M_{1}^{\prime }M_{1_{z}}-M_{2}^{\prime }M_{2_{z}}}{2}+\frac{L}{4}\cdot \left(\sin\left(\delta_{p}\right)-\sin\left(\delta_{r}\right)\right) \end{array}\right] \end{array}$$

As we have defined the normal vector $\vec{n}$ of the top plane *H*, the top plane can be described as the point set lying in the top plane *H* below:(7)$$H=\left\{x\in R^{3}:nx^{T}=x_{0}x_{1}+y_{0}y_{1}+z_{0}z_{1}\right\}$$

Based on the characteristics of this point set, the mid-points $M_{1}$ and $M_{2}$ are just lying on the top plane, so both of them are included in the set, and satisfy the equations below,
(8)$$\begin{array}{c} x_{0}\cdot O_{1}M_{1_{x}}+y_{0}\cdot O_{1}M_{1_{y}}+z_{0}\cdot O_{1}M_{1_{z}}\\ =x_{0}x_{1}+y_{0}y_{1}+z_{0}z_{1} \end{array}$$
(9)$$\begin{array}{c} x_{0}\cdot O_{1}M_{2_{x}}+y_{0}\cdot O_{1}M_{2_{y}}+z_{0}\cdot O_{1}M_{2_{z}}\\ =x_{0}x_{1}+y_{0}y_{1}+z_{0}z_{1} \end{array}$$ Based on the equations from (1) to (9), the kinematic related parameters are included from the input parameters, such as $\vec{M_{1}^{\prime }M_{1}}$ and $\vec{M_{2}^{\prime }M_{2}}$, to the output parameters, such as the Cardan angles $\delta_{r}$, $\delta_{p}$ and coordinate values $x_{0}$, $y_{0}$, $z_{0}$. Solving the equations above in MATLAB, we can have the inverse kinematic equations of this origami structure shown below,
(10)$$\begin{array}{cc} ∥\vec{M_{1}^{\prime }M_{1}}∥ & =(Lx_{0}\cos\left(\delta_{p}\right)-Lx_{0}-Ly_{0})\\ \; & +\frac{-L\sin(\delta_{p})+2r}{2}+\frac{r}{z_{0}} \end{array}$$
(11)$$\begin{array}{cc} ∥\vec{M_{2}^{\prime }M_{2}}∥ & =(Ly_{0}\cos\left(\delta_{r}\right)-Lx_{0}-Ly_{0})\\ \; & +\frac{-L\sin(\delta_{r})+2r}{2}+\frac{r}{z_{0}} \end{array}$$

From the derivation above, when the target position of this origami structure is set, we can determine the lengths of $∥\vec{M_{1}^{\prime }M_{1}}∥$ and $∥\vec{M_{2}^{\prime }M_{2}}∥$. Since these two values depend on the pneumatic input, we can realize the motion control through inputting the appropriate pneumatic pressure according to the actual calibration value.

#### 3.1.2. Mode Analysis of Origami Structure

In this subsection, we review the modal analysis of this foldable structure to verify the multiple motion modes to meet the requirements to realize various force-sensing types. Here, we only focus on this origami structure to accurately show all possible motion modes. The model has no external workload in this case.

From the results of the modal analysis shown in Figure 6, when actuated with pressurized air by the pneumatic actuation, the linear motion of the foldable structure compresses along the *Z* direction shown in Figure 6a, which also indicates that the maximum displacement is around 30 mm. When actuated by the torque along the transverse axis, the origami structure will generate the motion of pitch and roll accordingly, such as in Figure 6b,c. Its maximum roll or pitch angle can reach around $40^{\circ }$; and the diagonal motion, such as in Figure 6d, will be generated by applying the combined torque along the orthogonal transverse axes. The array motion shown in Figure 6e demonstrates that the flexibility of the origami structure can help sense the array force. Therefore, based on the motion analysis results above, we further verify that this foldable structure holds the capability to achieve multi-DOFs motions for force-sensing and can collect enough force information by combining the five motion modes above.

#### 3.1.3. Adaptation Test on Origami Structure

A further explanation of this experimental setup can verify the versatility of the origami structure’s motion; further experiments on the adaptation tests were carried out. The experiment setup in Figure 7a can be mainly made of a triaxial linear platform and a motor-driven rotary platform (Y200RA200). Two PLA holders from 3D printing are respectively attached to the linear platform and the rotary platform via nuts to fix the origami structure. The compression motion of the origami structure is achieved by moving the linear platform, while the rotary platform helps achieve the origami structure’s pitch, roll, and diagonal motions. The demonstration results in Figure 7b–d show that the maximum compression of the origami structure is around 27 mm, which is closer to the theoretical compression value—30 mm in the simulation analysis. However, the maximum left deflection angle of the pitch or roll motion is 21°, and the maximum right deflection angle is 23°. The maximum deflection angle of the diagonal motion is about 15°, which shows that the errors of the origami structure in achieving the bending motion are relatively higher compared to the simulation results. According to the analysis, the error is mainly caused by the thickness of the laminate structure, which further limits the deflection of this origami structure to a theoretical deflection angle of 40°. The diagonal motion is more significantly affected by the wall thickness, so the deflection angle is much smaller than that in the case of the pitch or roll motion. However, the origami structure successfully performs linear, pitch, roll, and diagonal motions. Therefore, as analyzed above, we can also demonstrate that this origami structure allows sensing multi-degree-of-freedom forces and has good adaptability.

### 3.2. Microfiber Sensor Analysis and Experiment Test

#### 3.2.1. Stress Distribution Analysis of Microfiber Sensor

This subsection presents a finite element mechanical simulation of the microfiber sensor to highlight the direct forces on the microfluidic tube more intuitively, allowing a direct comparison between the commonly used sandwich structure and the dual-layer structure proposed in this paper. Just as the following simulation presents [33] and based on our prior work on the micro-fiber sensor [34], the fixed constraint is added at the sensor’s bottom. The surface compression is applied to the surface of the microfiber sensor. Under the compression load condition, the maximum displacement values in the sandwich structure are distributed around the boundary of the upper layer. In contrast, those in the dual-layer structure are directly located on both ends of the micro-tubes, as shown in Figure 8a. Moreover, the maximum stress values are almost distributed on the micro-tubes in the dual-layer structure, as shown in Figure 8b. Therefore, the displacement distribution and the stress distribution chart results in Figure 8 show that the upper layer polymer will hinder the microfiber sensor from detecting the real pressure signal from the external compression. Through our design, the pressure will be directly applied to the microfiber sensor, alleviating the problem and improving the sensitivity of pressure awareness.

#### 3.2.2. Mechanical Test Setup

The mechanical properties of the microfiber sensor are characterized by an Instron 5943 Testing System (Norwood, MA, USA). The basic setup of a mechanical test is shown in Figure 9a. The circle load applying the schematic of the Instron tester is shown in Figure 9b. The actuator method is set as compression (speed: 0.1 mm/s), and the indenter with the same area size and shape is chosen for applying external compression. To validate the repeatability and stability of our proposed sensor, the cycling test is conducted in this experiment (cycling number: 10).

#### 3.2.3. Stability Test

In this test, repeated compression tests over ten cycles with peak pressures of 10 N, 20 N, 100 N, and 200 N were performed. According to the test results (shown in Figure 10a–d), the output signals exhibit no drift or fluctuation during the cyclic tests. Here, it is noted that the sensor peak resistance is stable under a low compression force. However, the peak resistance slightly varies under the large compression force because the stress distribution may change uniformly in the sensor and the bottom Eco-Flex layer during each loading. The potential capability of sensing an extensive of forces could be demonstrated. Therefore, the results above prove that our sensor obtains excellent stability under a relatively low compression force and wide force-sensing range.

#### 3.2.4. Calibration Test

In Figure 10e, we depict a relationship curve between the measured resistance value and externally applied pressure. The calibration curve shows the two-stage process in the resistance value versus the pressure curve. During the indentation, the sensor’s deformation causes a decrease in the cross-sectional area of the sensing element, and the microfiber’s resistance value increases. When the compression pressure is low, the soft layer deforms together with the microfiber sensor, which causes the sensor’s slight deformation. However, when the compression pressure increases, more stress is applied to the sensor itself, which would cause a more significant deformation. Therefore, The measurement range is divided into the approximate linear sensitivity zone (0–100 kPa) and exponential sensitivity zone (over 100 kPa). Here, the zoom view in the low-pressure region (0–100 kPa) is also represented, which indicates this sensor holds the relatively stable sensitivity of ∼0.35 Ω/kPa. In comparison, the microfiber sensor’s sensitivity exhibits an outstanding high value over a wide pressure range up to 350 kPa, which renders our device capable of working under various conditions.

### 3.3. Sensing System Experiment

In this section, the simplified experiment is reviewed to verify the feasibility of our system. The origami actuator is driven by pressurized air, and the microfiber sensor is embedded seamlessly into the origami structure to monitor the pressure in real-time. A customized circuit board with a Wi-Fi unit is used to control each air pump and solenoid valve to change the direction of the airflow, read the electric signal from the pressure sensor, and communicate with smart devices, such as computers, to transmit pressure data and receive commands. Following this working principle, the air path diagram according to this system is shown in Figure 11a. The experiment setup for pneumatic actuation is established in Figure 11b, and the origami actuator, valve, and pump are connected in series through air tubes. ESP32 MCU with a Wi-Fi antenna controls the valve and pump and then the origami motion. There are two working states for the origami actuator. The air is delivered into the device during the actuation state, and the origami structure is extended. According to the needs, the origami actuator holds the pressure for a specific time. When the treatment is finished, the air is pumped out and the pressure is released. Figure 11c,d shows the release state and actuation state. To better control the process of force rendering display, we programmed a control panel for our device. This interface receives the wireless data transmission by the ESP32 MCU. In the study, microfiber sensor resistance was measured using a Keithley DMM6500 digit multimeter, and the sensing data were saved directly into the computer in CSV files. To validate the feasibility of this force rendering device, the inner air pressure of the foldable structure with the constant value of 100.524 kPa is monitored by the commercial pressure sensor. Therefore, our microfiber sensor’s real-time resistance acquisition curve in Figure 11e can precisely record the change in the inner air pressure when the foldable state changes.

The final experiment is to verify that the pre-pressure value can affect the sensor sensitivity. The external pressure or calibration load is applied by the Instron system. The basic setup of the mechanical test is shown in Figure 11a. The actuator mode is set as compression (speed: 0.1 mm/s). The indenter embedded with the force sensor ATI Nano17-E is chosen to apply the external compression and record the force. Figure 12b indicates that with the increasing pre-pressure of 65–95 kPa, the average sensitivity curve shifts rightwards around 9 N intervals, which highly increases the force-sensing capability towards the range of 0–2 N. Figure 12c indicates that when the pre-pressure is at the relatively extreme air pressure 100 kPa, the force sensitivity value is around 11.6 Ω/N.

## 4. Conclusions

In this work, we propose an origami-inspired structure with variable stiffness to achieve multiple degrees of freedom (DoF) motions for force-sensing. The foldable structure is embedded with microfiber sensors through a multiple-layer integration fabrication. Finally, motion modal simulations and adaptability testing experiments demonstrate the flexibility of the origami-inspired structure. Additionally, the calibration and test experiments have successfully verified that the structure’s stiffness can be changed through pneumatic actuation to adjust the sensitivity of force-sensing. Although we verified that the origami sensing system could realize force-sensing, the experiments in our current work were conducted under ideal conditions without external loads.The additional contact loads are considered when the structure is in contact with the skin in a wearing situation. Therefore, further experiments on wearable sensing tests, and a more adaptive origami structure might be further promoted to increase its wearability and interactivity.

Moreover, the current modal analysis further demonstrated its flexibility due to combining the soft materials with the origami structure, and its ability to adapt to irregular and variable surfaces, such as the human skin surface. Therefore, the multi-mechanical-mode demonstration is more oriented towards passive force-sensing.While the perception of component forces along the x, y, and z directions are not considered in this paper, the current work only considers the change in sensitivity of the overall sensing system due to changes in pressure caused by external forces, so that only one sensing microfiber is introduced to capture the force information. However, it should be acknowledged that the introduction of several microfiber sensors will significantly enhance the ability of the current system to sense component forces from different directions. Meanwhile, the foldable structure design and the placement of the sensors might be thoroughly reconsidered, as will the decoupling of the component signals from each other. In the future, our foldable pressure-aware system can be embedded with multiple microfiber sensors and investigate the optimal parameters for body-mounted force rendering displays. More physical signals will be considered, such as angle and displacement measurements, and more functional components will be centralized in the system to detect human signals. Due to the low fabrication costs, a high integration level, and suitable sensing range, our wearable system can become affordable daily. It can be used at home according to the medical instructions from doctors.

## Figures and Tables

**Figure 1 sensors-22-05370-f001:**
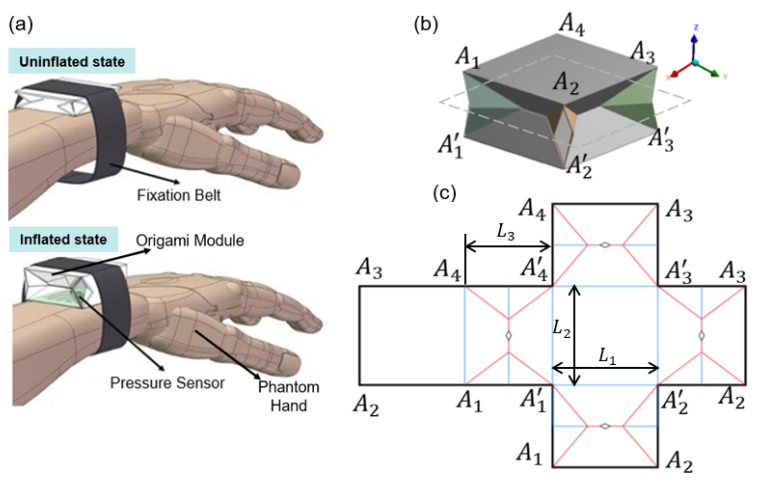
(**a**) The potential application scenarios of the wearable device for multiple DOF force-sensing: the foldable structure can be fixed to the fixed strap for easy wearing on the human wrist. The foldable structure in an uninflated state is easy to wear and can be adjusted to a fixed position on the wrist for wearing comfort; the foldable structure in an inflated state can be fixed on the wrist, and the sensitivity of the sensor can be variable by adjusting the pre-pressure value. (**b**) This core origami model is made of a parallel waterbomb-like unit. Based on the advantages of soft materials, the structure has multiple pseudo-DOFs compared to the rigid ones, allowing for various basic motion modes. (**c**) The folding pattern of this origami structure is shown, where solid red lines represent the mountain folds, and solid blue lines represent the valley folds. The essential size parameters $L_{1}$, $L_{2}$, and $L_{3}$ of this structure are designed as 60, 60, and 50 mm respectively. The diamond cutouts release the fixed constraints from the middle creases and stress concentrations from the folding process.

**Figure 2 sensors-22-05370-f002:**
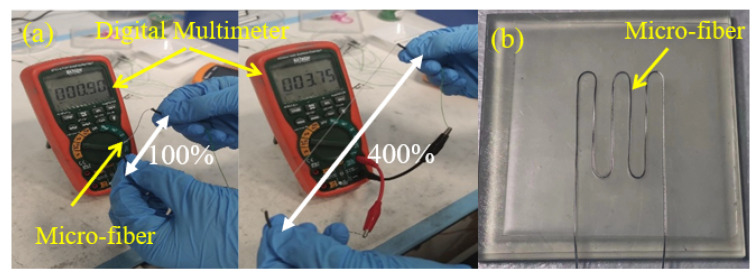
(**a**) Stretching test of liquid metal microfiber from 100% (**left**) to 400% (**right**), and the digital multimeter is utilized to measure the change of the resistance value; (**b**) shows the basic demonstration of the sensor layout.

**Figure 3 sensors-22-05370-f003:**
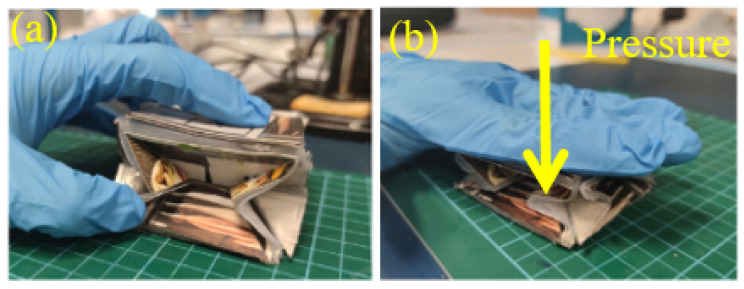
The passively linear motion of our origami structure. (**a**,**b**) show that this origami structure generates the linear motion when normal pressure is manually applied.

**Figure 4 sensors-22-05370-f004:**
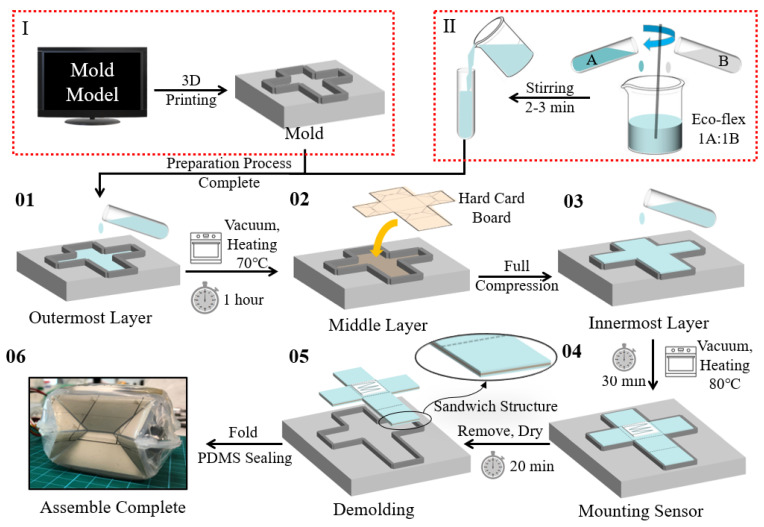
The fabrication process of the pressure-aware foldable structure can be mainly divided into the preparation process and layered the curing process. The preparation process includes I. The molding fabrication is by 3D printing and II. The Eco-Flex mixing solution preparation. After the preparation, the layered curing processes follow the steps: 01. Molding the outermost layer molding. 02. Attaching the middle layer. 03. Molding the innermost layer. 04. Mounting the proposed microfiber sensor. 05. De-molding the model of the sandwich structure. 06. Folding and PDMS, sealing the structure. To accelerate curing and compactness between layers, the molding procedure is processed in the vacuum oven with a temperature of 70 ${}^{\circ }$C to realize accelerated curing and bubble removal.

**Figure 5 sensors-22-05370-f005:**
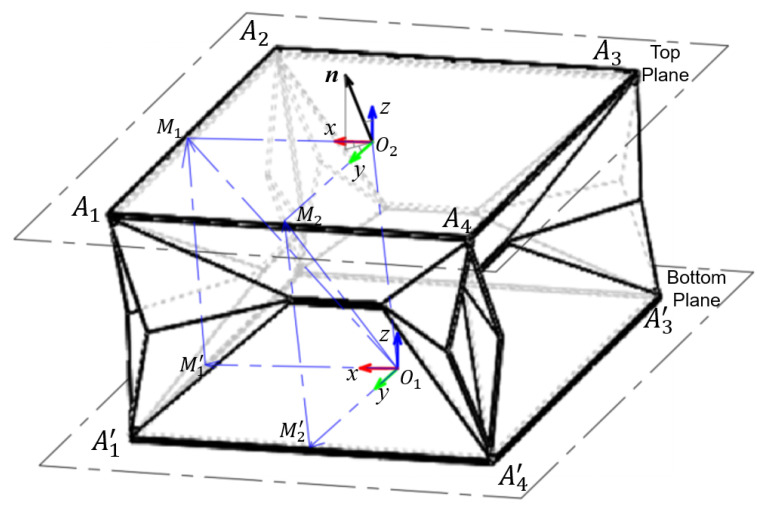
Origami kinematic model: we model our system as two rigid plates connected with four legs. When this foldable structure generates the pitch and roll motion, the top surface will form a certain angle relative to the bottom plane.

**Figure 6 sensors-22-05370-f006:**
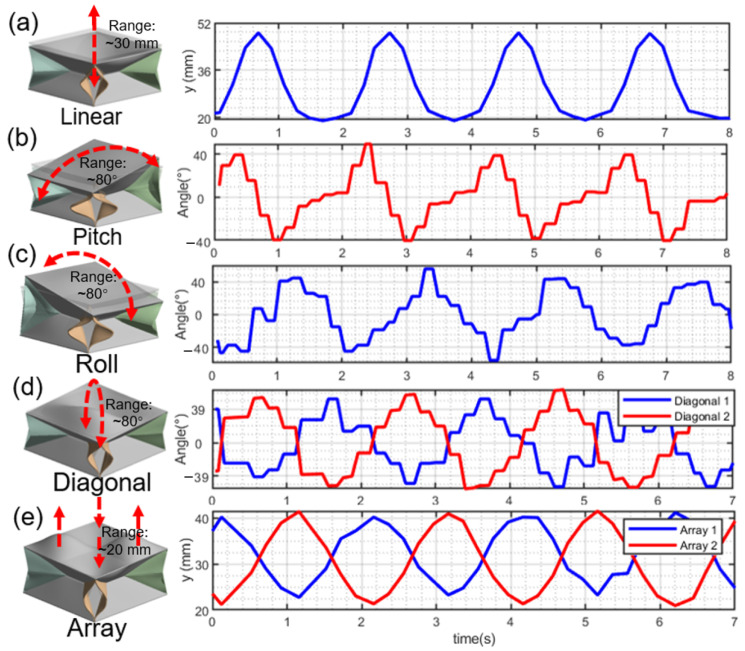
The five motion modes (linear, pitch, roll, diagonal, and array motion) of our origami structure are demonstrated from (**a**–**e**) based on the simulation results in ANSYS Workbench. Here, one vertex of origami structure is selected as the tracking objective, which indicates that the maximum displacement along the *Z* direction is around 30 mm, and the maximum pitch or roll angle is around $40^{\circ }$.

**Figure 7 sensors-22-05370-f007:**
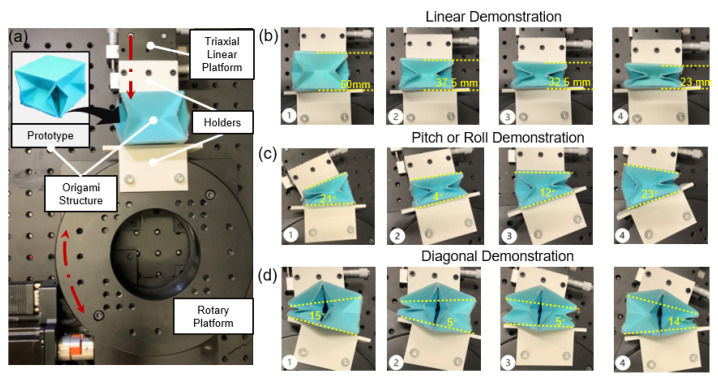
(**a**) Adaptation test setup. The experiment setup for the origami structure’s adaptation test. (**b**) The origami structure’s linear demonstration results. (**c**) The origami structure’s pitch or roll demonstration results. (**d**) The origami structure’s diagonal demonstration results.

**Figure 8 sensors-22-05370-f008:**
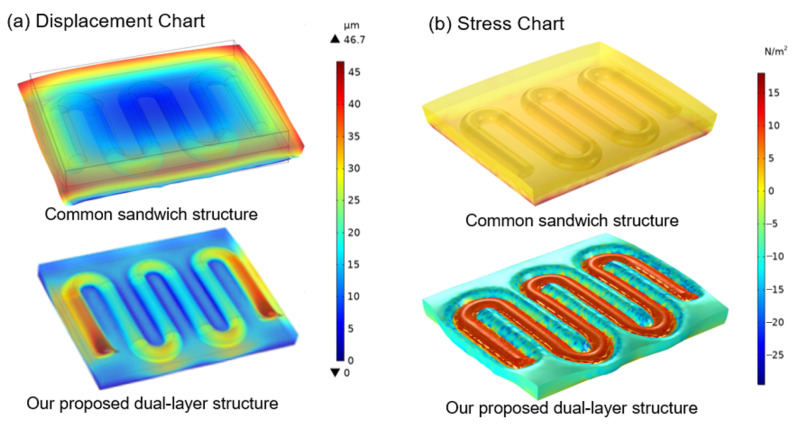
Comparison of simulation results on the (**a**) displacement chart and (**b**) stress chart for the microfiber sensor between the common sandwich structure with the upper layer and our proposed dual-layer structure without the upper layer under external pressure.

**Figure 9 sensors-22-05370-f009:**
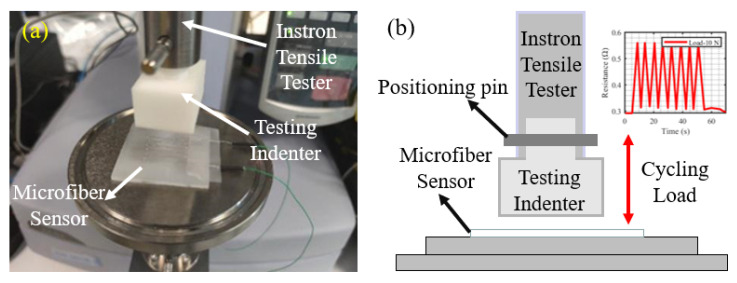
(**a**) The basic test setup system for the Instron tensile tester, which includes the testing indenter by 3D printing and our proposed microfiber sensor; (**b**) the working schematic of the Instron tester for the cycle load testing.

**Figure 10 sensors-22-05370-f010:**
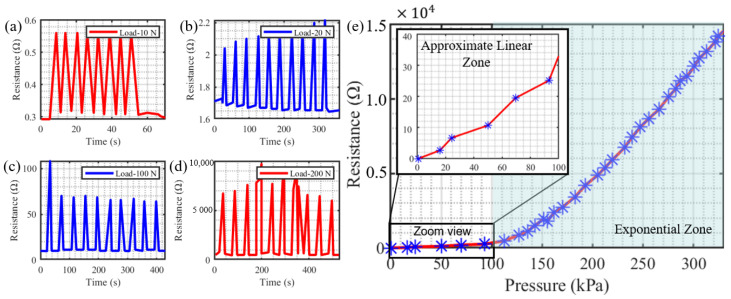
(**a**–**d**) indicate the working stability test curve: under–over 10 cycles for loads of 10 N, 20 N, 100 N, and 200 N respectively, the output signals exhibit excellent repeatability and stability; (**e**) refers to the calibration curve to describe the relationship between the resistance value and the pressure applied, and the measurement range is divided into two zones: approximate linear sensitivity zone (0–100 kPa) and exponential sensitivity zone (over 100 kPa).

**Figure 11 sensors-22-05370-f011:**
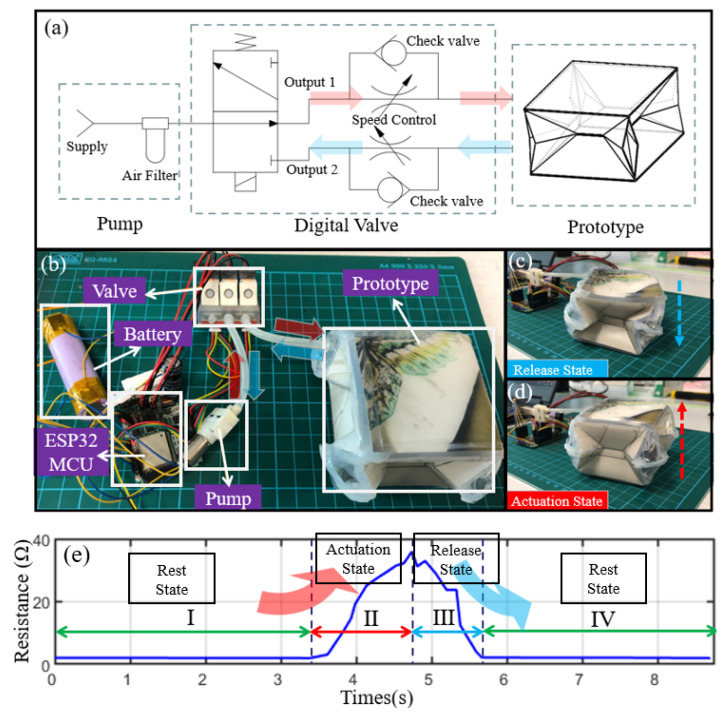
(**a**) Represents the air path diagram, where the whole system is divided into the air pump part, the digital valve controlled by the ESP32 MCU, and the foldable prototype. (**b**) There is a basic experimental setup for the pneumatic actuation of our prototype, and the valve is controlled by ESP32-MCU to realize the switch on intake or exhaust state; (**c**,**d**) indicate the two working states: the release state when exhausting and the actuation state when intaking. (**e**) The real-time resistance acquisition curve of our microfiber sensor can precisely record the air pressure change inside the foldable structure when intaking air with a constant pressure of 100.524 kPa. The foldable structure’s state is changed from I. Rest State; II. Actuation State; III. Release State; and finally back to the IV. Rest State.

**Figure 12 sensors-22-05370-f012:**
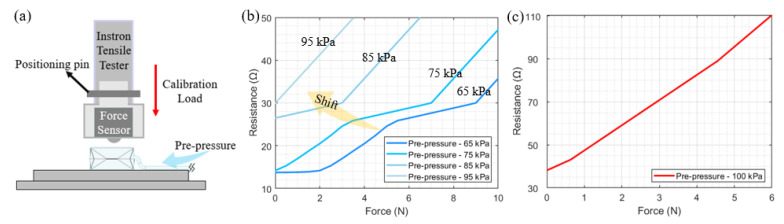
(**a**) Indicates the setup of the calibration experiment, and the force sensor is fixed into the suitable clamps (fabricated by 3D printing) to identify the load value; (**b**) represents the offset relationship curves between the microfiber sensor’s resistance and adding force when the pre-pressure is at the approximate linear range (65–95 kPa); (**c**) represents the relationship curves between the microfiber sensor’s resistance and adding force when the pre-pressure is at the exponential zone (100 kPa).

## Data Availability

Not applicable.

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
