# Peer review of "Origami-Inspired Structure with Pneumatic-Induced Variable Stiffness for Multi-DOF Force-Sensing"

_sensors, 2022, doi:10.3390/s22145370_

Round 1

Reviewer 1 Report

The authors present a manuscript describing results from a study on a origami-inspired structure where a strain micro-fibre sensor has been integrated. Modelling, several mechanical tests and measurements has been carried out. The result is interesting, but the manuscript needs to be revised. Comments are given below.

Addresses: check address affiliations of the authors.

Abstract: it is good to add what the resistance sensitivity means in terms of displacements or angles.

Figure 4: Several questions; a) add the figure numbers (01 - 06) also in the figure text so it is easier to follow the steps., b) It is not easy to see how the strain sensor micro-fibre is integrated in the structure. Add more about this, c) Is it only one strain microfiber that is integrated? 

Row 211 page 8: Multiple motion modes? but it is only one sensor signal (resistance in this case) since it is only one integrated sensor micro-fibre (or is it more than one sensor fibres?). See also comments below (conclusions).

Figure 6 page 9: add the displacement y and angle in the figures to the left so it is easier to read.

Figure 7 page 10: is it the micro-fibre sensor that can be seen in the figure? add that in the text. Add also some numbers to the colour bar (meaning the numbers of the strain).

Figure 9 page 11: Four comments. a) Is this right understood; A force of x N is applied and then the force is released and then x N is applied again, where x is 10 N, 20 N, ..., and the resistance is measured, is this right? What can be seen from figure a - d there are concerns about the sensor signal stability since the peak resistance (and base line) changes every time the same force is applied and released. Explain this. b) How is the pressure defined in figure e, what is the area being used to calculate the pressure (P=F/A). c) It looks also that there is a two-stage process in the Resistance versus Pressure curve (figure e, at low and high pressures) is it possible to add more about this. d) Is it the previous multimeter (shown in figure 2) that was used to measure the resistance here? How was the measured resistance data digitally transferred (DAQ-card). Just a comment; there are other more accurate instruments to measure small resistance changes (compared to a ordinary multimeter) and it is easy to transfer the data over to a computer.

Conclusion page 13: Even if the manuscript is interesting to read, the major concern about this sensor method is that the authors claims that this device can be used in a multi-mechanical-mode application but there is only one sensor micro-fibre that gives only one sensor signal of the force/strain. There must be several different mechanical modes on the origami-inspired-device that can give the same measured resistance from the micro-fibre. Is it possible to add several sensor micro-fibres to the device?

Author Response

Dear Reviewer,

Many thanks for your advices! It is very helpful to promote the quality of the article.

Reviewer 2 Report

The author presents an experimental approach to fabricate a werable force sensor that consists of origami structure and fiber-based sensor. Steps in the paper shows in-depth analysis and fabrication process. Although the pumping system are far from practicality, I applaud the demonstration. However, I suggest major revision for publishing paper in the MDPI sensor (under IF 30% in Eng. field).

1. The structure of the sensor has unique and large dimensions compare to other commercial items and sensor research. Thus, the author could provide appropriate user scenes what reders can recognize it's practicallity.

2. A force-sensing resistor, a significant comparer, has slim and flexible structures and has been well known for force sensing. Thus, the author should compare with it or provide how to overcome functionality of the most generous sensor.

3. Although the title significantly proposes multi-DOF sensing, the paper employs only simulation for emphatic actuation modes. The author could provide experimental supports.

4. The microfiber sensor is the one of three summary. However, there is a lack of data such as 'ultra stretching - electrical signal table', variation of simulation, Etc.

5. Author should add details for Figure 1(a). Plus, I suggest re-write the first caption according to balance of figures 1.

6. Add enough arrows and text on figures and plots. Reader can't understand images, such as 'diamond cutouts'.

7. Author should presents reason of glue prefered.

8. The analysis of 3.1.3 seems arguable. Need more explanation on 3.1.3.

9. Overall, it's technically descriptive. Reinforce discussions as a good research paper.

Sincerely

Author Response

(The authors gave the same response as above.)

Round 2

Reviewer 1 Report

The authors have revised the manuscript according to the comments.

Reviewer 2 Report

1. Address - clearly done

2. Overall review - expecting future works